# Probiotics for Reduction of Examination Stress in Students (PRESS) study: A randomized, double-blind, placebo-controlled trial of the probiotic *Lacticaseibacillus rhamnosus* HN001

**Rebecca F. Slykerman**[1]*, **Eileen Li**[2], **Edwin A. Mitchell**[3]

**1** Department of Psychological Medicine, University of Auckland, Auckland, New Zealand, **2** A Better Start – National Science Challenge University of Auckland, New Zealand, **3** Department of Pediatrics, Child & Youth Health, University of Auckland, New Zealand

☯ These authors contributed equally to this work.
* r.slykerman@auckland.ac.nz

## Abstract

### Background

Studies suggest that bioactive compounds such as probiotics may positively influence psychological health. This study aimed to determine whether supplementation with the probiotic *Lacticaseibacillus rhamnosus* HN001 reduced stress and improve psychological wellbeing in university students sitting examinations.

### Methods

In this randomized, double-blind, placebo-controlled study, 483 undergraduate students received either the probiotic *L. rhamnosus* HN001, or placebo, daily during a university semester. Students completed measures of stress, anxiety, and psychological wellbeing at baseline and post-intervention before examinations. Mann Whitney U tests compared the change in psychological outcomes between groups.

### Results

Of the 483 students, 391 (81.0%) completed the post-intervention questions. There was no significant difference between the probiotic and placebo supplemented groups in psychological health outcomes. The COVID19 pandemic restrictions may have influenced the typical trajectory of stress leading up to examinations.

### Conclusion

We found no evidence of significant benefit of probiotics on the psychological health of university students. These findings highlight the challenges of conducting probiotic trials in

**Data Availability Statement:** All relevant data are within the paper and its Supporting information files.

**Funding:** This study received funding from Fonterra Cooperative Group Limited. Grant number 5000855. The funders had no role in study design, data collection and analysis, decision to publish, or preparation of the manuscript.

**Competing interests:** The authors have declared that no competing interests exist.

human populations where the potential for contextual factors such as COVID19 response, and participant adherence to the intervention may influence results.

## Introduction

The microbiota in the human gut are a colony of microbes (bacteria, viruses, and fungi) that play an essential role in physiological and biochemical processes in the body. The microbiota-gut-brain axis refers to multidirectional signaling pathways that communicate between the microbes in the gut and the central nervous system, including the hypothalamic-pituitary-adrenal (HPA) axis, which modulates stress response and the immune system, both pathways linked to the experience of stress, anxiety, and depression [1–3].

Disruption to the gut microbial balance influences the biochemical metabolites produced in the gut, thereby altering neurotransmitter synthesis and modulating regulation of the HPA axis [4]. Stress, in turn, can alter the balance of microbiota in the gut. A recent review concluded that although there is substantial evidence from preclinical studies showing the gut microbiota influence the physiological stress response, further work in human populations is needed to realize the potential benefits of positively influencing the gut microbiota to manage stress [5].

Promising preclinical studies using mice have suggested that probiotics (defined as live microorganisms that, when ingested in sufficient quantity, confer a health benefit) may improve the mental health of human participants by enhancing the gastrointestinal microbial environment. In one of the foundational studies in the field, germ-free mice, who have no commensal microbiota, displayed exaggerated responses to stress which improved with probiotic supplementation [6]. Understanding the therapeutic role of probiotics in humans is far less well elucidated, in part due to the incredible complexity of human physiology and psychology. A systematic review of the evidence for the benefit of probiotics on subclinical symptoms of psychological stress concluded that probiotic supplementation could reduce depression, anxiety, and perceived stress in healthy volunteers. However, small sample sizes were a limitation of most of these studies [7].

Conversely, in their systematic review and meta-analysis of 10 randomized controlled probiotic trials for depression and anxiety, Pirbaglou et al. (2016) suggested there was limited evidence that probiotics could reduce depression and anxiety. Six of the reviewed studies were in healthy volunteers. Of those, two studies found statistically significant improvements in depression or anxiety attributable to probiotic supplementation [8]. The need to translate preclinical research into therapeutic benefits for people with psychiatric or stress-related conditions remains [9, 10]. With the prevalence of stress and its impact on physical and psychological health, probiotic supplementation trials are critical to advancing the potential utility of probiotics in humans.

University students experience increased stress associated with examinations. A study of Japanese medical students found that stress increased over eight weeks before a national examination peaking the day before and then decreased to baseline two weeks later. The same study found that those students supplemented with *Lactobacillus casei* (strain Shirota) had fewer gastrointestinal, cold, or flu-like symptoms [11, 12]. A study of stressed American university students reported similar results, those supplemented with *Bifidobacterium bifidum* reported fewer days of cold and flu symptoms and more healthy days than those who received a placebo [13]. A previous study of 155 university students randomly allocated students to receive either

a multispecies probiotic containing *Lactobacillus casei DN 114001*, *Lactobacillus delbrueckii subsp*. *Bulgaricus*, and Streptococcus salivarius subsp. Thermophilus or placebo daily for six weeks which included three weeks leading up to examinations and three weeks during the examination period. They found that scores on the State Trait Anxiety Inventory increased for all students, but there were no significant differences between probiotic and placebo groups. However, the probiotic supplemented group showed a statistically significant increase in markers of immune function [14].

To date, there has not been a large study of the effect of probiotic supplementation for psychological symptoms of stress, anxiety, and mental wellbeing in university students. Previously, in the Probiotics in Pregnancy (PIP) Study, 423 pregnant women were randomly assigned to receive *L. rhamnosus* HN001 or placebo daily from enrolment at 14–16 weeks gestation until six months postpartum. Mothers supplemented with *Lactobacillus rhamnosus* HN001 had significantly lower postnatal depression and anxiety scores than the placebo group. Furthermore, supplementation with *L. rhamnosus* HN001 improved anxiety and depression in all women, not only those who had high baseline levels of depression or anxiety [15]. In addition, the HN001 strain was associated with improved mental wellbeing in a population of prediabetic adults during a calorie-restricted diet [16], suggesting that probiotic supplementation with the HN001 strain may be beneficial for psychological wellbeing. Furthermore, this benefit may confer to all participants, not simply those who already have high levels of stress, anxiety, or depression.

This study aimed to investigate whether supplementation with the probiotic *Lacticaseibacillus rhamnosus* HN001 reduced the build-up of stress, reduced symptoms of anxiety, and improved psychological wellbeing in university students leading up to examinations. We chose HN001 based on our previous work which demonstrated a positive effect on mood in women who were supplemented during pregnancy with this probiotic [15].

## Materials and methods

The study design was a randomized, double-blind, placebo-controlled trial with two parallel arms and an allocation ratio of 1:1 intervention to placebo.

### Participants

The New Zealand university system operates on two semesters per year, with examinations for a paper held at the end of a semester. Participants were undergraduate students at the University of Auckland enrolled in semester one of 2020. Exclusion criteria were, currently taking a regular probiotic supplement, taking immunosuppressants, e.g., chemotherapy, or current participation in another research trial. Participants were recruited between 9 March 2020 and 13 April 2020, through advertisement of the study in lectures, and by placement of study information on the online resource page for individual courses.

### Data collection

All consent and data were collected through an online web interface. Using their mobile phones, tablet, or computer, students registered to participate in the study between 9 March and 13 April 2020 and completed baseline questionnaires about psychological health at the time they registered. Students answered the same questionnaires at the end of the intervention period two days prior to the commencement of examinations between 15 June and 17 June 2020.

## Intervention

Fonterra Cooperative Group Limited supplied capsules containing the probiotic *Lacticaseibacillus rhamnosus* HN001 ($6 \times 10^9$ colony forming units) manufactured to pharmaceutical grade. Placebo capsules identical in appearance and smell to the probiotic contain corn-derived maltodextrin. Both probiotics and placebo capsules are lactose-free, gluten-free, and contain no animal products. Previous studies have safely used the probiotic *L. rhamnosus* HN001 ($6 \times 10^9$ cfu) in studies conducted in New Zealand, including in pregnant women [15] and infants [17].

Instructions to students were to take one capsule a day from when they enrolled in the study and received the capsules until two days before the commencement of university examinations for the semester. Participants received text messages reminding them to take the capsules at fortnightly intervals during the intervention period. Treatment time ranged from 8–13 weeks depending on when participants enrolled in the study.

## Randomisation

Randomisation was managed by Fonterra Co-Operative Group Limited and concealed from study staff and participants. A computer generated randomisation list was used to assign participants to receive either probiotic or placebo. As students completed baseline consent and registration information, they were assigned the following available sequential study number and provided with the corresponding bottle of capsules according to the study number.

## Outcome measures

The primary outcome of interest was self-reported symptoms of stress. Anxiety and psychological wellbeing were secondary outcomes.

**Stress.** The Perceived Stress Scale is a 10 item questionnaire that asks about stress and coping in the previous month. Scores range from 0–40, with higher scores being indicative of higher levels of stress. Scores from 0–13 represent low stress, 14–26 indicate moderate stress, and 27–40 indicate high stress.

**Anxiety.** The State Trait Anxiety Inventory 6 item version (STAI6) is a short 6 item scale validated as an anxiety screening questionnaire based on the more extended State Trait Anxiety Inventory [18, 19]. Clinically significant anxiety was defined as a score above a cut-off of score >15.

**Psychological well-Being.** The World Health Organisation—Five Well-Being Index (WHO-5) is a five-item, positively worded measure of psychological wellbeing with scores ranging from 0 to 25. Higher scores represent better wellbeing. Scores of 13 or lower indicate low levels of psychological wellbeing. A systematic review of the WHO-5 concluded that it was a widely used and sensitive measure of depression [20].

## Impact of COVID19

Semester one of 2020 teaching at the University of Auckland commenced on 2 March 2020. At that time international news was emerging detailing rising concern about the spread of COVID19, but no cases had been detected in New Zealand and there was minimal impact of the virus at a local level. Three weeks later, the University of Auckland instituted a teaching-free week to provide staff time to prepare for the increasing likelihood that on-campus teaching would be disrupted. At the beginning of the teaching-free week, on 23 March 2020, The New Zealand government announced lockdown restrictions that would require all people to stay at home with three exceptions: essential workers providing healthcare or food distribution

services could travel to and from their workplace, people could leave home to buy food, or to access medical care. Consequently, after three weeks of in-person teaching, the University of Auckland moved exclusively to a remote delivery model for the remainder of semester one, including online examinations at the end of the semester.

Because all data collection for this trial was managed using a web-based platform, recruitment of students continued, and capsules were sent to participants by courier.

### Sample size

The initial sample size for the trial was calculated based on a clinically meaningful reduction in stress of one quarter of a standard deviation in stress score. To give a 90% chance of detecting a difference of 1.6 points (0.25 SD) in Perceived Stress Scale scores between the probiotic supplemented group and placebo group at the 5% level of significance, 326 participants in each arm of the study would be required. This equals a sample size of 652. Allowing for 25% attrition over the intervention period, a total of 815 participants was the initial target sample size. The government-led restrictions to manage the spread of COVID19, and the resulting closure of the university campuses, meant that the initial sample size target of 815 students was not achieved.

The actual sample size achieved in this trial was 483 students of whom 391 completed end of intervention data collection. Therefore, our final sample size gave a 90% chance of detecting a difference in Perceived Stress Scale scores of 2.9 points or 0.45 SD between the probiotic and placebo groups, at the 5% level of significance.

### Statistical analysis

Intent-to-treat analysis was conducted in SAS 9.4 using Mann Whitney U test. Change in stress, anxiety, and psychological wellbeing was calculated by subtracting baseline scores from post-intervention scores for each of the three measures. The findings are reported according to the CONSORT statement.

## Results

Of the 483 participants initially enrolled in the trial, 391 (81.0%) completed the end-of-intervention questions. Fig 1 shows the CONSORT flow diagram for the trial. There was no significant difference between respondents and non-respondents to the end of intervention questions in the intervention group (p = 0.66), sex (p = 0.91), ethnicity (p = 0.51), study paper (p = 0.65), or year of study (p = 0.42). No adverse events were reported by participants during the study.

Table 1 shows the characteristics of the study sample. The probiotic supplemented, and placebo groups did not significantly differ in demographic factors or measures of psychological health.

Table 2 shows the outcome measure results according to study group allocation. There were no significant differences between the probiotic and placebo groups in median change in stress, anxiety, or psychological wellbeing.

## Discussion

In this randomized, double-blind, placebo-controlled trial of the probiotic *Lacticaseibacillus rhamnosus* HN001, we found no significant difference in stress, anxiety, and psychological wellbeing in university students between the placebo and probiotic intervention groups. Although there is promising evidence that probiotic supplementation may improve the

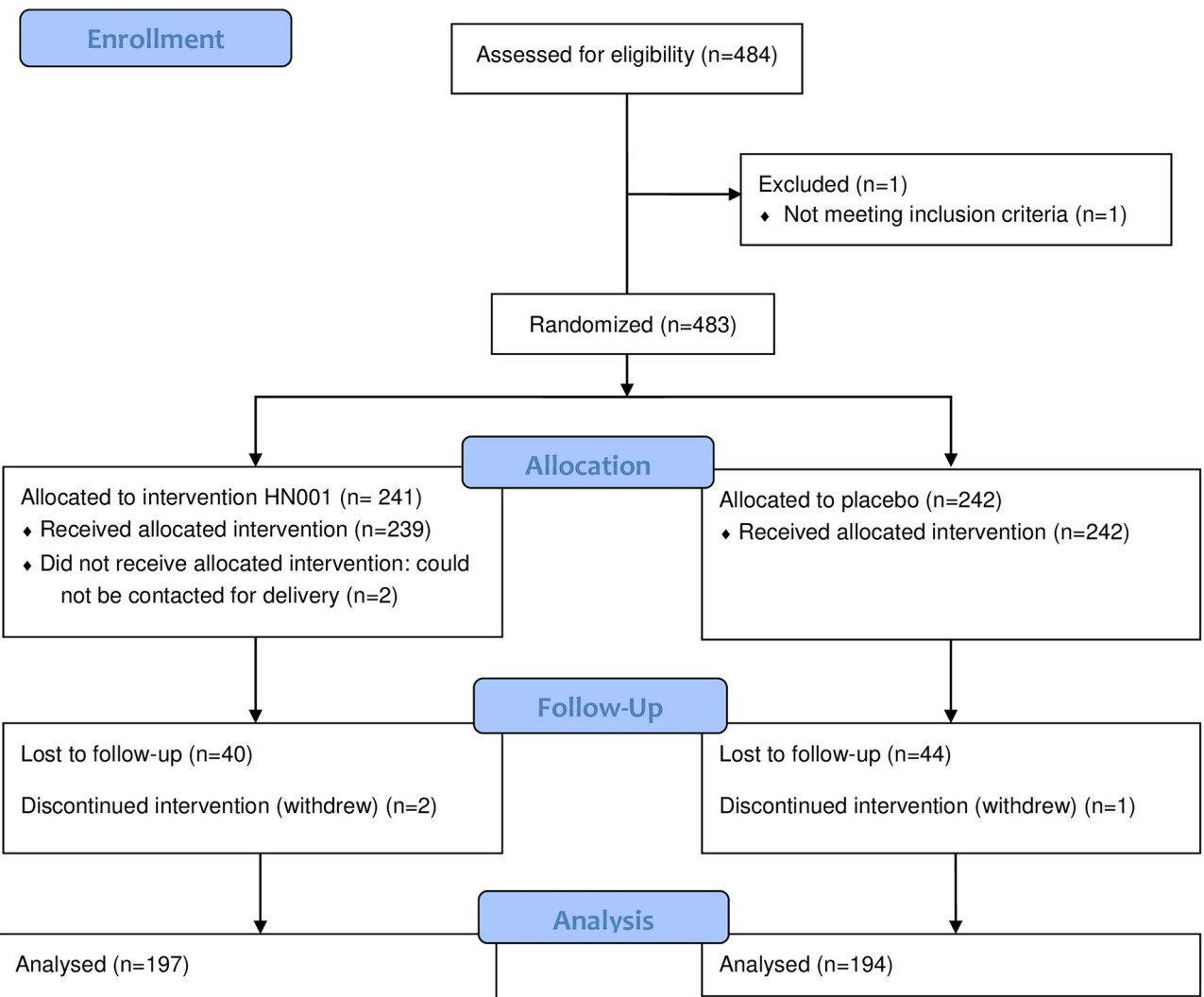

**Fig 1. CONSORT flow diagram showing recruitment and group assignment.**

stress-induced suppressed immune function in students, these previous trials have also not shown a significant difference in self-reported psychological symptoms of stress, anxiety, or depression in university participants [11–14]. In a previous trial of *L. rhamnosus* JB-1 in healthy volunteers, there was no significant difference between placebo and probiotic groups on stress, depression, and anxiety or physiological measures, including HPA-axis function, neurocognitive and inflammatory markers, despite preclinical evidence in a mouse model showing promising results for this strain of probiotic. This finding highlights the difficulties in translating promising preclinical evidence into human populations [21].

It is well understood in the probiotic field that benefits for health associated with probiotics are strain-specific [21, 22]. While we did not find evidence in the current study that the HN001 strain improved psychological outcomes where other strains may have, in the Probiotics in Pregnancy study we have previously demonstrated that the HN001 strain was associated with significantly lowered depression and anxiety scores in postpartum mothers [15]. Furthermore, a recent trial showed improvements in psychological wellbeing associated with HN001 probiotic supplementation in prediabetic adults on an intermittent fasting diet [16].

**Table 1. Participant characteristics.**

|  | N (%) |
|---|---|
| **Sex** |  |
| Male | 363 (75.16) |
| Female | 117 (24.22) |
| Unspecified | 3 (0.62) |
| **Ethnicity** |  |
| European | 216 (44.72) |
| Māaori | 38 (7.87) |
| Pacific | 20 (4.14) |
| Asian | 174 (36.02) |
| Other | 35 (7.25) |
| **Study Paper** |  |
| Medicine | 77 (16.63) |
| Medical Science | 13 (2.81) |
| Population Health | 180 (38.88) |
| Psychology | 145 (31.32) |
| Other | 48 (10.37) |
| **Study Year** |  |
| First year | 278 (60.04) |
| Second or third year | 185 (39.96) |
| **Baseline scores (n = 483)** | **Median (25th, 75th percentile)** |
| Stress | 21 (16, 25) |
| Anxiety | 53 (50, 57) |
| Wellbeing | 13 (10, 16) |
| **Post Intervention Scores (n = 391)** | **Median (25th, 75th percentile)** |
| Stress | 18 (14, 23) |
| Anxiety | 53 (50, 57) |
| Wellbeing | 13 (10, 17) |

The potential impact of COVID19 on this research requires discussion. The university semester this study was conducted in was interrupted by government-led restrictions that forced the closure of the university campus and a complete shift to online learning, studying, and assessment for the semester. The COVID19 pandemic is often assumed to increase stress; however, the opposite was true for our cohort of university students, and this may have influenced our study results. New Zealand has been one of the most successful countries worldwide in containing and eliminating COVID19. During the university semester of this study, New Zealand moved from Alert Level 4 (complete lockdown) to Alert Level 1 (no restrictions except on international travel). Students may have experienced a reduction in stress before their

**Table 2. Median change in stress, anxiety, and psychological wellbeing scores between baseline to post-intervention for the probiotic and placebo groups.**

|  | Placebo (n = 194) | HN001 (n = 197) | Pr > \|Z\| |
|---|---|---|---|
|  | Median change (25, 75 percentile) | | |
| Stress | 2 (-2, 6) | 2 (-2, 5) | 0.64 |
| Anxiety | 0 (-3, 6) | 0 (-3, 4) | 0.42 |
| Wellbeing | 0 (-3, 2) | 0 (-3, 2) | 0.83 |

examinations due to a significant improvement in eliminating COVID19 from the community that resulted in a shift down to Alert Level 1 during the semester.

New Zealand's lockdown was one of the most restrictive in the western world. Stay at home instructions asked people to restrict outings to essential personal movement, travel was severely limited, gatherings were canceled, and all public venues closed, businesses were closed except for essential services, and all educational facilities were closed. As a result of these closures which occurred early in the university semester, many students returned home where they may have received more social support from family and therefore experienced fewer symptoms of stress. In addition, examinations for students were online and uninvigilated, and this may also have resulted in a reduction of stress for students leading up to examinations in direct contrast to previously reported increase in pre-examination stress in studies conducted before COVID19 [12, 23]. Literature indicates that some students find online examinations less stressful [24, 25]. Lastly, the closure of the university campus impacted our rate of recruitment, and we were not able to reach our intended sample size of 815. Nevertheless, our sample size of 483 participants is relatively large for a randomised trial in the field of probiotic research and retention of participants at the end of intervention was high (81%). The failure to find a significant difference between groups (that is to reject the null hypothesis) does not exclude the possibility of a small benefit of probiotics. The introduction of measures to limit the spread of COVID19 resulted in a reduced sample size than planned and therefore we may not have detected the presence of a smaller benefit of probiotics for the psychological health of students.

A further possible explanation for the lack of a significant association between probiotic supplementation and psychological outcomes is that young adult university students in our sample may have had reduced adherence to the daily capsule intake. Adherence to medication is low in young adults compared with older age groups, even in the presence of chronic illness [26]. In addition, the COVID19 lockdown announcement in New Zealand occurred at the beginning of week four of the university semester. Many students left Auckland to return to other parts of New Zealand for the lockdown, and it is possible that this may have interrupted adherence to daily capsule intake. Total treatment time was a minimum of 8 weeks and up to 13 weeks depending on when participants enrolled in the study. It is expected that variation in intervention period would be similar in the probiotic and placebo groups given participants were randomly assigned to the intervention. It is possible that a longer length of intervention may have resulted in benefit for psychological health. However, this is less likely given previous studies have used intervention periods of 8 weeks or less [8].

These factors may limit the generalisability of our results to the wider population of university students in a typical university semester uninterrupted by campus closure. Despite this, our findings make an important contribution to the field of probiotic supplementation for mental wellbeing and highlight some of the challenges of conducting trials that can be influenced by wider contextual and societal factors.

## Conclusions

In this large double-blind, randomized placebo-controlled trial of the probiotic *L. rhamnosus* HN001 in university students, we found no significant difference between supplementation groups in stress, anxiety, or psychological wellbeing before end-of-semester examinations. The COVID19 elimination measures implemented during the study period may have impacted any potential benefit of probiotics as psychological wellbeing in the students improved contrary to previous reports that suggest wellbeing typically deteriorates before examinations. Our study further highlights the difficulty of realizing the potential benefit of probiotics for

different health outcomes and indicates that further research across populations and probiotic species is required.

## Supporting information

**S1 File. Deidentified dataset for the PRESS trial.** Deidentified dataset.
(DOCX)

**S1 Checklist. CONSORT 2010 checklist of information to include when reporting a randomised trial***.
(DOC)

**S1 Data.**
(XLSX)

## Acknowledgments

We would like to thank the academic and professional staff members from the University of Auckland who kindly assisted in recruiting students through announcements about the study. We would also like to thank the participating students who continued to be involved in the study during a semester affected by COVID19.

### Ethical approval

The study received full ethical approval from the University of Auckland Human Participants Ethics Committee: Reference 023964.

### Trial registration

The trial was prospectively registered with the Australian New Zealand Clinical Trials Registry ACTRN12620000275965.

## Author Contributions

**Conceptualization:** Rebecca F. Slykerman, Edwin A. Mitchell.

**Data curation:** Rebecca F. Slykerman, Eileen Li, Edwin A. Mitchell.

**Formal analysis:** Eileen Li.

**Funding acquisition:** Rebecca F. Slykerman, Edwin A. Mitchell.

**Investigation:** Rebecca F. Slykerman.

**Methodology:** Edwin A. Mitchell.

**Project administration:** Rebecca F. Slykerman.

**Writing – original draft:** Rebecca F. Slykerman.

**Writing – review & editing:** Rebecca F. Slykerman, Eileen Li, Edwin A. Mitchell.

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
