## [Decision Letter · Decision Letter 0]

23 Dec 2021

PONE-D-21-34328Probiotics for Reduction of Examination Stress in Students (PRESS) Study: A randomized, double-blind, placebo-controlled trial of the probiotic Lactobacillus rhamnosus HN001PLOS ONE

Dear Dr. Slykerman,

Thank you for submitting your manuscript to PLOS ONE. After careful consideration, we feel that it has merit but does not fully meet PLOS ONE’s publication criteria as it currently stands. Therefore, we invite you to submit a revised version of the manuscript that addresses the points raised during the review process. There are some major statistical issues raise by one the reviewers that must be taken into account. Additionally, the issue of treatment compliance should be adequately addressed. 

We look forward to receiving your revised manuscript.

Kind regards,

Felipe Dal Pizzol

Academic Editor

PLOS ONE

Journal Requirements:

(This study was funded by Fonterra Cooperative Group Limited)

(This project received funding from Fonterra Cooperative Group Limited.)

5. Thank you for submitting the above manuscript to PLOS ONE. During our internal evaluation of the manuscript, we found significant text overlap between your submission and the following previously published work, of which you are an author.

- https://www.ANZCTR.org.au/Steps11and12/379312-(Uploaded-20-02-2020-07-02-22)-Study-related%20document.docx

Please revise the manuscript to rephrase the duplicated text, cite your sources, and provide details as to how the current manuscript advances on previous work. Please note that further consideration is dependent on the submission of a manuscript that addresses these concerns about the overlap in text with published work.

Reviewers' comments:

Reviewer's Responses to Questions

**Comments to the Author**

1. Is the manuscript technically sound, and do the data support the conclusions?

Reviewer #1: Partly

Reviewer #2: Partly

2. Has the statistical analysis been performed appropriately and rigorously? 

Reviewer #1: Yes

Reviewer #2: No

3. Have the authors made all data underlying the findings in their manuscript fully available?

Reviewer #1: Yes

Reviewer #2: Yes

4. Is the manuscript presented in an intelligible fashion and written in standard English?

Reviewer #1: No

Reviewer #2: Yes

5. Review Comments to the Author

Reviewer #1: The clinical trial developed by Slykerman et al is relevant, but I have some concerns.

It's unclear why the authors chose the Rhamnosus strain. Since in the introduction the authors report benefits of several probiotic strains.

In my opinion there is a concern about adherence to treatment by students.

How can authors be sure students take the capsule daily?

Treatment time is unclear too. How many days did the students take the capsules? Time is a determining factor in the result.

In conclusion authors state: "difficulty of translating preclinical evidence". However, the authors cite several clinical trials showing the strain's efficacy for anxiety and stress (see references 15, 16 and 7 and 8).

I think adherence to treatment and time can interfere with the results.

Reviewer #2: Important note: This review pertains only to ‘statistical aspects’ of the study and so ‘clinical aspects’ [like medical importance, relevance of the study, ‘clinical significance and implication(s)’ of the whole study, etc.] are to be evaluated [should be assessed] separately/independently. Further please note that any ‘statistical review’ is generally done under the assumption that (such) study specific methodological [as well as execution] issues are perfectly taken care of by the investigator(s). This review is not an exception to that and so does not cover clinical aspects {however, seldom comments are made only if those issues are intimately / scientifically related & intermingle with ‘statistical aspects’ of the study}. Agreed that ‘statistical methods’ are used as just tools here, however, they are vital part of methodology [and so should be given due importance]. To improve the article/presentation, clues/hints may be taken from this review but should not limit the process by adhering to those points alone.

COMMENTS: Though the measures/tools used are appropriate [like Perceived Stress Scale, State Trait Anxiety Inventory, World Health Organisation - Five Well-Being Index] all/most of them yield data that are in [at the most] ‘ordinal’ level of measurement [and not in ratio level of measurement for sure {as the score two times higher does not indicate presence of that parameter/phenomenon as double (for example, a Visual Analogue Scales VAS score or say ‘depression’ score)}]. Then application of suitable non-parametric test(s) is/are indicated/advisable [even if distribution may be ‘Gaussian’ (i.e. normal)]. Agreed that there is/are no non-parametric test(s)/technique(s) available to be used as alternative in all situation(s) [suitable / most desired/applicable], but should be used whenever/wherever they are available. According to lines 32-33 {T-tests compared the change in psychological outcomes between groups} using parametric test(s) is not indicated/desired/correct. That may not change the direction of the result(s); however, correct methods are to be used always.

Moreover, remember that “Absence of evidence is not evidence of absence” [Altman DG, Bland JM. BMJ volume 311, 1995, p 485 (Reprinted : Australian Veterinary Journal 1996;74, 311)]. {Even when P-value is not significantly lower that is null hypothesis of no difference is not rejected, (in short, result is not significant), that does not amount to evidence of absence i.e. it does imply that there no difference. It only implies that there is no (i.e. these samples do not provide) enough evidence to prove (rather indicate with certain specified confidence level) the difference}.

Refer to lines 254 onwards [‘Explanation for the lack of a significant association between probiotic supplementation and psychological outcomes is that young adult university students in our sample may have had reduced adherence to the daily capsule intake], may not suffice. As said, adherence to medication is low in young adults compared with older age groups, even in the presence of chronic illness [26], may be true but trialist is/are expected to take steps/precautions to avoid or reduce this phenomenon {as for this manuscript ‘Article Type: Clinical Trial’ and not observational study}. Such excuse is not welcome or expected with respect a clinical trial in academic circles, in my experience.

Further, note that to provide a description of baseline characteristics is entirely reasonable (since it is clearly important in assessing to whom the results of the trial can be applied), however, it does not require the division of baseline characteristics by treatment groups (however, if done – alright). Statistical comparison of baseline characteristics [last ‘p-value’ column in Table 1: Participant characteristics by intervention group] is not desirable at all [because even if P-value turns out to be significant (while comparing baseline characteristics despite random allocation), it is, by definition, a false positive] as you then are supposed to be testing ‘randomization’ then, which in any single trial may not balance all baseline characteristics because ‘randomization’ is a sort of ‘insurance’ and not a guarantee scheme.

References:

1. Stuart J. Pocock, et al., ‘Subgroup analysis, covariate adjustment and baseline comparisons in clinical trial reporting: current practice and problems’, Statistics in medicine, 2002; 21:2917–2930 [Particularly page 2927]

2. Harrington D, et al., ‘New guidelines for statistical reporting in the journal’, N Engl J Med 2019;381:285-6

[Important message (indirectly/ultimately indicated) from these articles: Never do any comparison with respect to ‘baseline’ characteristics {by applying statistical significance test(s)}, when allocation is done randomly].

In fact, comparison with respect to ‘Post Intervention Scores’ should not be done unless you adjust for ‘baseline’ [may be ANCOVA]. ‘Change scores’ {Table 2: Mean change in stress, anxiety and psychological wellbeing scores between baseline to post-intervention for the probiotic and placebo groups} may be tested for significance by suitable ‘non-parametric’ test [say Mann-Whitney ‘U’ test].

Information given in lines 169-177 regarding the ‘Sample size’ is not as per the CONSORT guidelines. This is {our final sample size gave a 90% chance of detecting a difference in Perceived Stress Scale scores of 2.9 points between the probiotic and placebo groups, at the 5% level of significance} not the way to give the information required, in my opinion. What is required is it [required sample size for the study] was ‘estimated’ (namely the assumptions).

In my considered opinion, ‘let the respected editor decide the future course’.

6. PLOS authors have the option to publish the peer review history of their article (what does this mean?). If published, this will include your full peer review and any attached files.

Reviewer #1: **Yes: **Monique Michels

Reviewer #2: No

---

## [Author Response · Author response to Decision Letter 0]

1 Mar 2022

Thank you for the opportunity to revise our manuscript in light of comments from the Reviewers. We have carefully considered each point and made changes to the manuscript to address each comment. We have uploaded an itemised response to review which details changes made and responses to each comment raised.

---

## [Decision Letter · Decision Letter 1]

18 Apr 2022

Probiotics for Reduction of Examination Stress in Students (PRESS) Study: A randomized, double-blind, placebo-controlled trial of the probiotic Lactobacillus rhamnosus HN001

PONE-D-21-34328R1

Dear Dr. Slykerman,

We’re pleased to inform you that your manuscript has been judged scientifically suitable for publication and will be formally accepted for publication once it meets all outstanding technical requirements.

Kind regards,

Felipe Dal Pizzol

Academic Editor

PLOS ONE

Additional Editor Comments (optional):

Reviewers' comments:

Reviewer's Responses to Questions

**Comments to the Author**

1. If the authors have adequately addressed your comments raised in a previous round of review and you feel that this manuscript is now acceptable for publication, you may indicate that here to bypass the “Comments to the Author” section, enter your conflict of interest statement in the “Confidential to Editor” section, and submit your "Accept" recommendation.

Reviewer #1: All comments have been addressed

Reviewer #2: All comments have been addressed

2. Is the manuscript technically sound, and do the data support the conclusions?

Reviewer #1: Yes

Reviewer #2: (No Response)

3. Has the statistical analysis been performed appropriately and rigorously? 

Reviewer #1: Yes

Reviewer #2: (No Response)

4. Have the authors made all data underlying the findings in their manuscript fully available?

Reviewer #1: Yes

Reviewer #2: (No Response)

5. Is the manuscript presented in an intelligible fashion and written in standard English?

Reviewer #1: Yes

Reviewer #2: (No Response)

6. Review Comments to the Author

Reviewer #1: Authors have adequately addressed all comments. The manuscript is now suitable for publication. Accept

Reviewer #2: COMMENTS: Since all of the comments made on earlier draft by me (and hopefully by other respected reviewers also) were/are attended positively, I recommend the acceptance because the manuscript now has achieved acceptable level, in my opinion.

Nevertheless, I wish/want to let authors note that, even if [while the Non-parametric test(s) is/are used] along with ‘median’ and ‘inter-q’ range, ‘mean’ and ‘SD’ can be given for ‘description’ as they are more common and describe well (help distinguish groups).

7. PLOS authors have the option to publish the peer review history of their article (what does this mean?). If published, this will include your full peer review and any attached files.

Reviewer #1: **Yes: **Monique Michels

Reviewer #2: **Yes: **Dr. Sanjeev Sarmukaddam

---

## [Editor Report · Acceptance letter]

12 May 2022

PONE-D-21-34328R1 

Probiotics for Reduction of Examination Stress in Students (PRESS) Study: A randomized, double-blind, placebo-controlled trial of the probiotic *Lacticaseibacillus rhamnosus* HN001 

Dear Dr. Slykerman:

I'm pleased to inform you that your manuscript has been deemed suitable for publication in PLOS ONE. Congratulations! Your manuscript is now with our production department. 

Kind regards, 

on behalf of

Dr. Felipe Dal Pizzol 

Academic Editor

PLOS ONE